# Identification of Parameters and Fatigue Life Assessment of the Road Pavement Lower Construction Layers under Heavy Construction Traffic

**DOI:** 10.3390/ma15165646

**Published:** 2022-08-17

**Authors:** Piotr Mackiewicz, Bartłomiej Krawczyk

**Affiliations:** Faculty of Civil Engineering, Wrocław University of Science and Technology, 50-370 Wrocław, Poland

**Keywords:** pavement, fatigue, subgrade, modules, FEM, light weight deflectometer

## Abstract

This article analyzes the results of testing the subgrade and the lower layers of the pavement structure with the light weight deflectometer at a load of 0.1 and 0.15 MPa. It is shown that, with layer systems with an equivalent layer modulus lower than 80 MPa, significant nonlinear phenomena occur at a load of 0.15 MPa. In this situation, the identification of a reliable replacement module, a commonly used test method, at a load of 0.1 MPa, is not appropriate—it significantly overestimates the value of the modules (even by 34%), which in turn translates into a significant overestimation of the fatigue life of the structure. In a situation where intensive exploitation of the lower layers of the pavement structure is planned before the final layer arrangement is made, it is required to apply test loads corresponding to the stress conditions occurring in these layers of the structure. Such a situation takes place under the influence of technological (construction) or temporary traffic (substitute, e.g., by-pass) during construction. In order to verify the above assumptions, numerical calculations (FEM) were carried out in the elastic model for layered structures with replacement modules determined in field tests. It was found that, especially in the case of low-bearing layer systems, it is necessary to use correction factors for modules determined with a dynamic plate at a load of 0.1 MPa. Taking into account the corrected values of the modules will allow to correctly determine the change in the durability of layers at the construction stage and in the subsequent operation of the final pavement structure.

## 1. Introduction

The assessment of the bearing capacity and compaction parameters of the pavement structure base with the use of the light weight deflectometer (LWD) is an issue that is still relevant in world road technology [1,2,3,4,5,6]. The dynamic plate is a device known and used for many years and, thanks to its efficiency, it gradually replaces the time-consuming static methods. However, the problem at present is the correct interpretation of the results of measurements of surface displacement (deflection) with a dynamic plate. In recent years, research was carried out on the identification of subgrade parameters using the light weight deflectometer and comparative studies with other applied research methods [7,8,9]. So far, they have not brought an unequivocal solution to the problem of interpretation of the results obtained in the dynamic plate test. Many authors analyzed phenomena occurring during dynamic loading test. Empirical correlations were made [10,11,12] between static and dynamic tests, and numerical analyses were conducted [13,14,15,16] for different types of subgrades. Additional sensors (geophones) [17] were also used to compare the LWD and the FWD (falling weight deflectometer) tests results [18,19]. A lot of effort is put into finding the correlation between the static and dynamic moduli for different subgrades [20,21,22,23,24] and different coefficients of subgrade reaction [25].

The test with an LWD consists of registering vertical displacements (deflections) of the subgrade under the loading impulse caused by a freely falling mass. A detailed description of the LWD is presented later in the article. The vertical displacements recorded in the test under a known load are the basis for further identification of the ground parameters. It is a complicated issue because the parameters of the layer to be assessed depend on its thickness, moisture, and the stiffness of the layers underneath. It is assumed that the impact range of a light dynamic deflectometer does not exceed 0.5 m, and it depends on the diameter of the loading plate and the value of the loading impulse.

There is a belief in the world road literature that testing with a light dynamic deflectometer, despite remaining ambiguities, is the most effective method, surpassing others and worth further improvement. The loading impulse generated in the LWD test is very similar to the character of the wheel load of the passing vehicle, and the stresses appearing in the ground correspond to the actual stresses occurring during the subsequent operation of the pavement structure. Different diameters of pressure plates are used around the world, with different discharge heights and sizes of the freely falling mass, which results in different values of the loading impulse (stresses at the contact of the pressure plate with the surface). The tests showed that the value of the identified parameters (stiffness modules) of the layer depends largely on the value of the impulse and the diameter of the plate, as well as the cyclical nature of the load (change of the stiffness modules at successive drops) [26]. Commonly, for practical reasons, simplified methods of assessing layer parameters based on a single value of the instantaneous maximum displacement (deflection) are used, without the knowledge and use of the entire deflection time course. They can lead to significant errors in identifying the subgrade parameters and underestimating the durability of the exploited layers at the construction stage.

In construction practice, it is very common for the lower layers of the structure, and even the improved subgrade itself, to be subjected to intensive technological or temporary traffic loads before the final arrangement of the pavement structure layers is made. As a result of stresses greater than those present in the target structure, the lower layers undergo earlier degradation, which reduces the durability of the target structure. In this case, it is very important to correctly assess the parameters of the lower layers of the pavement and subgrade structure, under the stress conditions occurring in the layers at the stage of their operation. The change in the durability of the lower layers of the pavement structure as a result of intensive use at the construction stage is not taken into account in road design practice, and may be very important—in recent years, accelerated degradation of the final structure was observed in areas where the lower layers were previously intensively used with construction, technological or temporary traffic.

The aim of this article is to show that, in the layer systems with equivalent modulus of 80 MPa or lower, significant nonlinear phenomena occur, when under stress of 0.15 MPa (identified in the model as the average). Therefore, the identification of the equivalent modulus should be conducted under such a stress. Identification under lower stress (0.1 MPa) is incorrect and causes significant overestimation of the equivalent modulus and, consequently, and overestimation of the fatigue life of the pavement construction.

## 2. Research Methodology

In the next stage of this publication, the results of tests with a dynamic plate at a load of 0.1 MPa and 0.15 MPa are presented on three partially constructed pavement structures, loaded with technological (construction) and temporary traffic (in the case of structures No. 1 and 2).

−construction No. 1: crushed stone 25 cm on a layer of cement-stabilized sandy loam 15 cm, underneath sandy loam subgrade;−construction No. 2: 15 cm layer of cement-stabilized sandy loam, underneath sandy loam subgrade;−structure No. 3: compacted native soil—compacted sandy loam subgrade.

Figure 1 shows the test stands and diagrams of the tested structures.

The test with a light dynamic plate (LWD) consists of inducing the appropriate stress in the tested layer and recording the response—displacement within about 20 ms. The mass freely falling from a predetermined height hits the damping system, which deforms and smoothly transmits the load impulse to the pressure plate. The plate, in turn, causes a vertical displacement (deflection) of the subgrade, which is registered by a special sensor, called a geophone. More details on geophone operation can be found in [27]. Two plates with a diameter of 0.3 m were used for the tests, differing in the size of the freely falling mass and the drop height. In the case of the first plate, the pressure on the subgrade was 0.1 MPa, and in the case of the second—0.15 MPa.

In each of the experimental fields, three measurements of vertical displacements with a dynamic LWD plate were carried out, according to the standard procedure [28], each time performing six discharges—loading cycles (three initial and three basic), and the results of all discharges (including three preliminary) were recorded. Figure 2 shows the results of measurements of vertical displacements depending on the number of loading cycles.

On the basis of the presented measurement results, it is possible to see the influence of the number of load cycles on the recorded vertical displacements in the LWD test, and it depends on the type of subgrade. It is most noticeable in the case of the sandy loam subgrade, and less in the case of crushed stone, on the stabilization layer. In the case of the native subgrade, the largest scatter of results was also obtained, which was visualized in the form of standard deviation. 

In the case of the compacted native subgrade (structure 3), at the load of 0.15 MPa, the final six discharge displacements obtained over three times greater displacements than for the cement-stabilized layer on the native subgrade (structure 2) and seven times greater than on the crushed stone layer (structure 1). It is worth noting that the obtained displacement values relate to the displacement values at the moment of the maximum load impulse. Figure 3 presents examples of displacement waveforms during the test for all six load cycles.

It should be noted that the crushed stone (sandwich system with the highest stiffness) has the smallest material damping effect, which is visible in the form of a large change in displacements after unloading. Lower damping, for all sandwich systems, is visible for a higher load of 0.15 MPa. 

Based on the comparison of the ratio of the mean values of displacements for the load of 0.15 MPa and 0.1 MPa (marked as y0.15/y0.1), it was found that sandwich system No. 1 with the crushed aggregate layer is similar to the behavior of the elastic material. The ratio of the displacement values oscillates around the value of 1.5, which corresponds to the ratio of the load values (Figure 4). A clear lack of proportionality of displacements in relation to the load is visible for the compacted native subgrade (sandy loam) for the final test cycle and amounts to 2.03. The value of 1.77 was obtained on the native subgrade, stabilized with cement (system No. 2). The same value was obtained for both the first and the last cycle, which proves the low variability of elastic properties with increasing load cycles. The lack of obtaining exact ratios equal to 1.5 proves the non-linear behavior of these materials (especially the compacted native subgrade) and indicates the need to use load-dependent material characteristics for their description or to use the correction coefficients developed by the authors.

The discussed material characteristics, determined on the basis of the displacement values, in relation to the load (for final discharge, which corresponds to a compacted layer system), are shown in Figure 5.

The recorded displacement results were used in the further part of the work to identify the values of dynamic equivalent moduli (*Evd*) of the examined layer systems. Figure 6 shows the values of dynamic moduli determined for the initial and final load cycles at different load values. The dynamic modulus *Evd* was calculated on the basis of the measured values of displacements *y_i_*, depending on the value of the load *σ_i_*:*Evd_i_* = π × (1 − *ν*^2^) × *σ_i_* × *a*/2 × *y_i_*,(1)
where: 

*Evd*—dynamic surface modulus [MPa],

*ν*—Poisson’s Ratio, 0.35 [-],

*σ_i_*—maximum contact stress [MPa]

*a*—plate radius = 150 [mm], 

*y_i_*—maximum deflection [mm], 

*i* for load 0.1 MPa or 0.15 MPa.

It is clearly visible how important the change of the modulus value may be in relation to the first and the last (sixth) load cycle, for which the registered displacements are already “stabilized”. Both in the case of the crushed stone layer on the cement-stabilized layer and in the case of the compacted native subgrade, the difference between the modules is almost twofold. Most likely, it results from the significant compaction of the material (in the case of stone in system No. 1) and plastic deformation of the material (in the case of sandy loam: No. 3). The smallest changes occur for system No. 2—the cement-stabilized native subgrade. It should also be noted that, for a plate load of 0.15 MPa, lower modulus values were obtained than for a load of 0.1 MPa. In the case of the analysis for the end cycles, the differences amount to, respectively, for the system No. 1, 4%; No. 2; 16%; and No. 3, as much as 35%. Such changes significantly confirm the “non-linear” material properties and indicate the need to correct the target values of modules if they will be tested under a load lower than the actual load that will occur during operation. 

Figure 7 shows the dependence of the displacement ratios at two load levels (0.1 MPa and 0.15 MPa) on the dynamic modulus determined at the load of 0.1 MPa. Of course, a clear correlation is noticeable for the final, stabilized load cycle. This implies the need to determine the correction factor k for the dynamic modules determined in the test using a plate load of 0.1 MPa (Figure 8). The determined dependence shown in Figure 8 allows for determining reliable corrected modules that can be used to estimate the durability of the exploited pavement layers in the case of a load greater than 0.1 MPa.

If the expected load from construction or temporary traffic causes stress of 0.15 MPa in the layers (such stress is identified in the model as the average occurring in the layer), the modules should be tested under the load of 0.15 MPa, or reduction factors for the designated modules, in the case of plate testing 0.1 MPa, should be applied. Otherwise, oversized systems with incorrectly determined layer parameters will degrade faster than the fatigue criteria indicates. In order to explain these phenomena more precisely and to determine the differences in fatigue life, numerical calculations were carried out for the examined layer systems.

## 3. Numerical Modeling

The finite element method (FEM) was used to calculate the value of displacements for different values of material parameters. The model was built with the use of three-dimensional volumetric elements (Figure 9), and the subgrade layers were described for comparison with elastic parameters corresponding to dynamic replacement modules determined in field tests. The load was assumed following the conditions of training tests—as a model of a rigid round plate with a pressure of 0.1 MPa and 0.15 MPa, and a radius of 0.15 m. Full interlayer adhesion was applied in the model.

In the description of the elastic material, a wide range of modulus changes from 5 MPa to 200 MPa was analyzed. All analyses were performed for the conditions in the final (sixth), stabilized load cycle. The FEM calculations were conducted for the following moduli obtained in tests: crushed stone on a cement-stabilized subgrade (construction No. 1): 139 MPa (load 0.1 MPa), 134 MPa (load 0.15 MPa); cement-stabilized subgrade (construction No. 2): 50 MPa (load 0.1 MPa), 42 MPa (load 0.15 MPa); native subgrade (construction No. 3): 30 MPa (load 0.1 MPa), 15 MPa (load 0.15 MPa). Poisson’s ratio of 0.35 was used in all cases.

## 4. Durability Analysis

Based on field studies and numerical calculations, comparative analyses were carried out (Figure 10). The compliance of the determined displacements in numerical calculations with the use of dynamic load was found in comparison with displacements obtained in field tests. Compliance occurs for both 0.1 MPa and 0.15 MPa load. The differences in values did not exceed 10%.

It should be noted that, for moduli with a value lower than 50 MPa, the displacement values clearly increase, which will significantly affect the durability of systems with weaker parameters. The structural deformation criterion was used to assess the pavement durability.

The criterion allows to determine the durability of the structure to the creation of a critical structural deformation of 12.5 mm, based on the relationship between the permissible number of repetitive loads *N* and the vertical deformation of the subsoil *ε_p_*:*ε_p_* = 1.05 × 10^−2^ × (1/*N*)^0.223^,(2)

Preliminary calculations of the deformation values were carried out for the tested sandwich systems with the use of FEM, in the previously described model, at the load of 0.1 MPa and 0.15 MPa. 

Figure 11, Figure 12 and Figure 13 show the values of the determined vertical deformations for the systems tested in the field. Up to a depth of about 20 cm, there are still relatively large deformations, constituting 70% of the maximum deformations under the edges of the slab. At a depth of about 40 cm, vertical deformations account for 40% of the maximum.

Due to the fact that the lower layers of the pavement structure are subject to technological (construction) or temporary traffic with direct pressure of 0.7 MPa, additional durability calculations were carried out for higher loads than the LPD slab. Figure 14 shows the dependence of the pavement durability on the value of equivalent-surface moduli and different load levels.

The durability calculations show that the analyzed systems with modules less than 50 MPa may not transfer the required construction or temporary traffic and will be destroyed in the upper part of the layer when in contact with a pressure of 0.7 MPa. At a depth of about 20 cm, the values of vertical stresses will decrease to about 0.3 MPa, which ensures durability at this depth of about 100 axes. Such layers will therefore require partial replacement with new layers. In the case of other, more rigid systems (e.g., with cement-bound layers), a reduction in durability should be taken into account in the comprehensive operation of the final pavement structure, depending on the intensity of technological/temporary traffic. The durability of these layers at different depths will be from 100 to 10,000 axes.

## 5. Stress Analysis in Pavement Layers

In order to assess the durability of the structure as a result of deformation of the subsoil, an analysis of various pavement structures made and operated in stages during construction was carried out. In the calculations, the FEM model was used, taking into account the appropriate layer arrangement depending on a given case. The calculations were carried out for the vehicle wheel load with a radius of 0.16 m and a pressure of 0.71 MPa, and the systems of elastic layers with the following modules: 10,000 MPa asphalt concrete; 400 MPa crushed stone; 250 MPa cement-stabilized soil (sandy loam); and 30 MPa native subgrade (sandy loam).

Three stages were considered, which in practice are often made and used during construction with technological (construction) or temporary traffic. Stage 1: subgrade improved from native cement-stabilized native soil (0.15 m). Stage 2: an improved stratified subgrade, made of crushed stone (0.25 m) on a cement-stabilized native subgrade (0.15 m). Stage 3: as Stage 2, but with an additional layer of asphalt concrete (0.14 m)—the final structure.

For the diagrams, vertical stresses in the successively built layers of the target pavement structure were analyzed. Figure 15 shows the distribution of vertical stresses in layer systems at individual construction stages. 

The gray field indicates the minimum stress range of 0.1 MPa, which appears in the key structure layer, which may be subject to accelerated degradation. The following layers were analyzed: cement-stabilized native soil and crushed stone since, for these layers, the criterion of vertical deformation can be used in assessing their durability. Large values (above 0.1 MPa) of stress values, firstly, adversely affect the degradation of the layers, even before the entire structure is completed; and secondly, they indicate that the assessment of compaction and identification of the replacement module of the layers should be performed with a load greater than the commonly used 0.1 MPa. 

It Is worth noting that, in the case of stage 1, the stresses in the subgrade are apparently low at 0.17 MPa on average, but they cause high micro-deformations 5420, which translates into the durability of only 19 axes. For stage 2 (with an additional layer of stone), stresses of 0.04 MPa are present in the subgrade, ensuring durability for the subgrade 13,001 axes. In the aggregate layer, despite the high level of stresses and, at the same time, the high value of the modulus, relatively small micro-deformations (697) occur, which ensure the durability of the axis for this layer. In the final structure (stage 3), the durability of the subgrade is 1,848,994 axes, and the aggregate layers are 75,180,511 axes. The comparison of deformation values and subgrade durability for different stages is shown in Figure 16.

The differences in durability are very large. The subgrade, even in the case of a cement-stabilized layer, is not able to transfer the technological movement. A degraded subgrade with a low modulus will not be able to transfer the next loads after building subsequent layers (i.e., stage 2). For example, when the subgrade modulus drops to 10 MPa, its durability will be 543 axes, and the durability of the aggregate layer will decrease by almost 50%. In stage 3, reduced durability of the structure can be expected, which will decrease by a factor of 10. The stabilized layer will also undoubtedly be degraded. Due to the complex process of degradation of the bound layers, this issue is not discussed in this publication. The authors only wish to draw attention to the important fact of the need to conduct tests for loads above 0.1 MPa. It is of key importance to include dynamic tests already at the construction stage and to identify replacement modules with increased loading of the layers. This fact should also be taken into account at the design stage with the calculation of durability at individual stages and further use of the pavement. Particular attention should be paid to the bottom layer of stabilization and crushed stone. The difference in the modules of these layers (400 MPa vs. 250 MPa) is about 1.6, which obviously translates into the values of replacement modules, determined for systems containing these layers at different load values (0.1 MPa vs. 0.15 MPa). As a result of the authors’ research with plates with different values of the loading impulse, it is shown that the difference is from a few to several percent.

## 6. Conclusions

In this article, the authors demonstrate the necessity to adjust the test loads in the light weight deflectometer test to the stress conditions occurring in the earlier exploitation of the lower layers of the pavement and subgrade, loaded with technological (construction) or temporary traffic. Inadequate test load leads to incorrect identification of modules and, consequently, overestimated fatigue life of the used layers and the final pavement structure. Due to the non-linear behavior of the lower layers of pavement and subsoil structures under load, it is necessary to use the load-dependent material characteristics for their description or to use correction factors under a test load of 0.1 MPa. Due to the significant degradation of these layers under a load of technological (construction) or temporary traffic (before the implementation of the final layer system), it is necessary to rationally plan the construction, staging, and operation of the layers of the pavement and subgrade structure, before the implementation of the target system, with simultaneous control of the degree of their degradation (replacement of layer parameters identified in tests with a dynamic plate). The reduction in the fatigue life of the lower layers of the pavement and subgrade structure should be taken into account at the design stage of the structure in the future operation of the target system of the pavement structure layers.

Special attention should be paid to fatigue life analysis of the low bearing subgrades under temporary or construction traffic. In such cases, the correction factors should be used when identifying moduli in an LWD test under 0.1 MPa load. It is essential to correctly predict the fatigue life in the following stages of the construction and of the final layer system.

## Figures and Tables

**Figure 1 materials-15-05646-f001:**
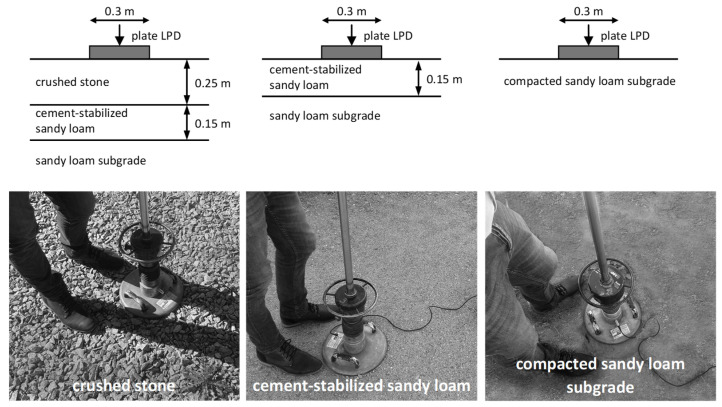
Test stands and diagrams of the tested structures.

**Figure 2 materials-15-05646-f002:**
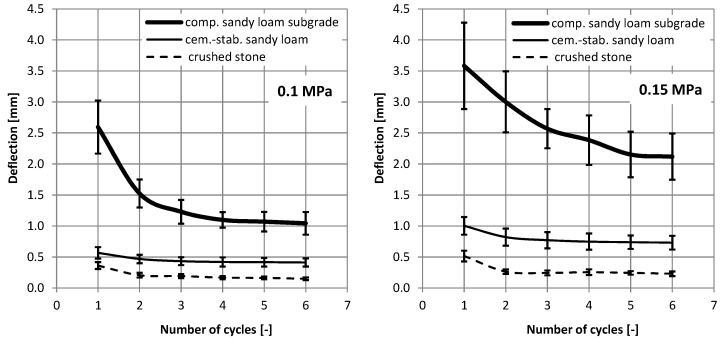
Vertical displacements as a function of the number of load cycles, depending on the type of the tested subgrade at the load of 0.1 and 0.15 MPa.

**Figure 3 materials-15-05646-f003:**
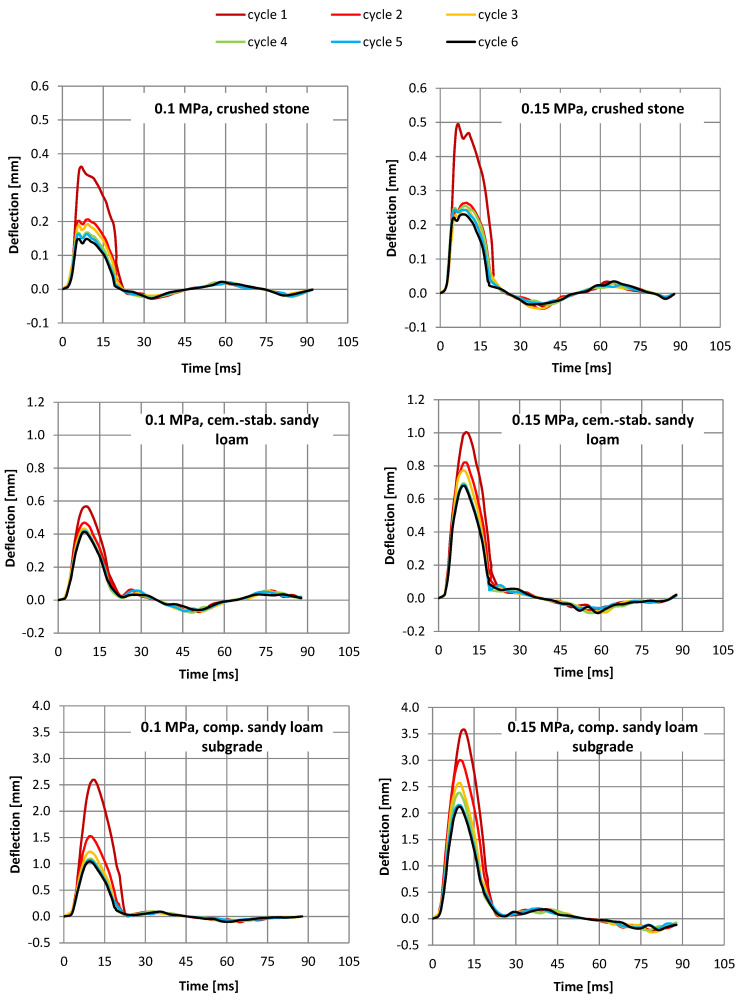
Time courses of displacements during the test with a dynamic plate, depending on the tested layer, at a load of 0.1 and 0.15 MPa.

**Figure 4 materials-15-05646-f004:**
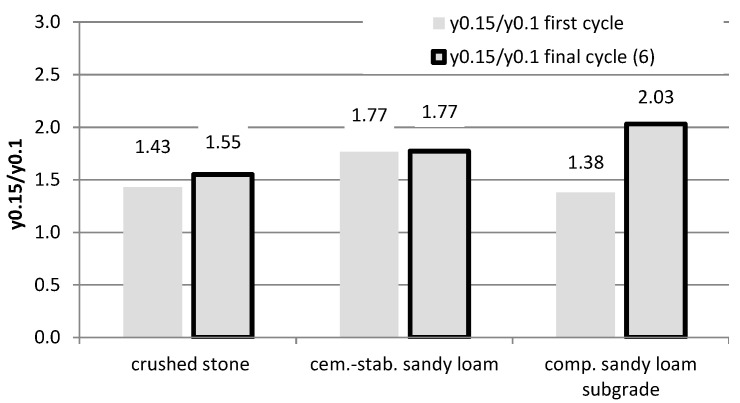
Ratios of displacements obtained in the plate test under the load of 0.15 MPa and 0.1 MPa.

**Figure 5 materials-15-05646-f005:**
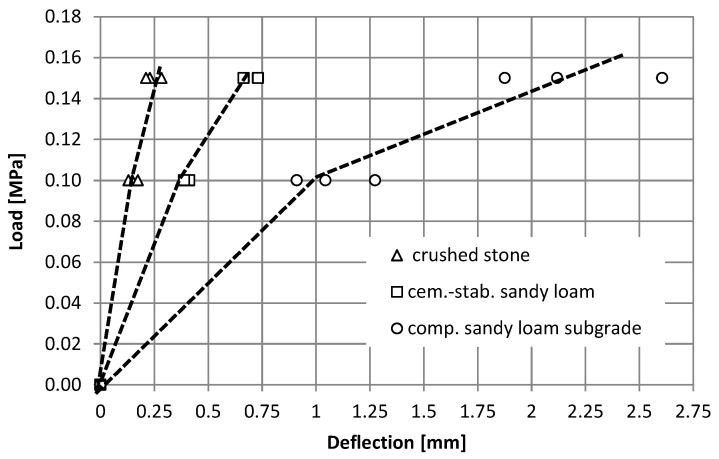
Material characteristics of the tested pavements depending on the load value (final cycle).

**Figure 6 materials-15-05646-f006:**
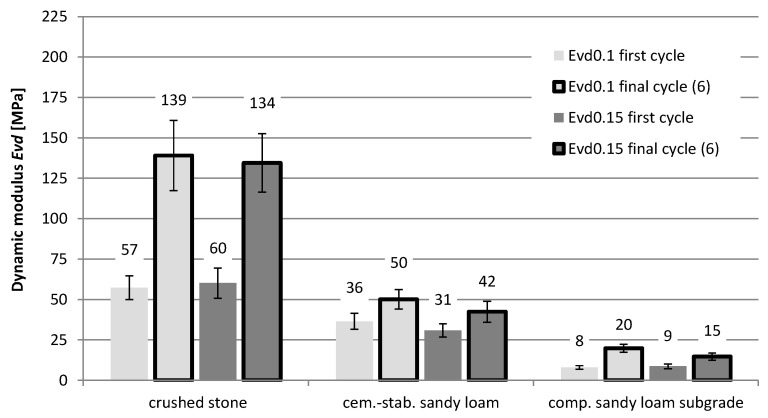
Values of dynamic moduli of the tested structures.

**Figure 7 materials-15-05646-f007:**
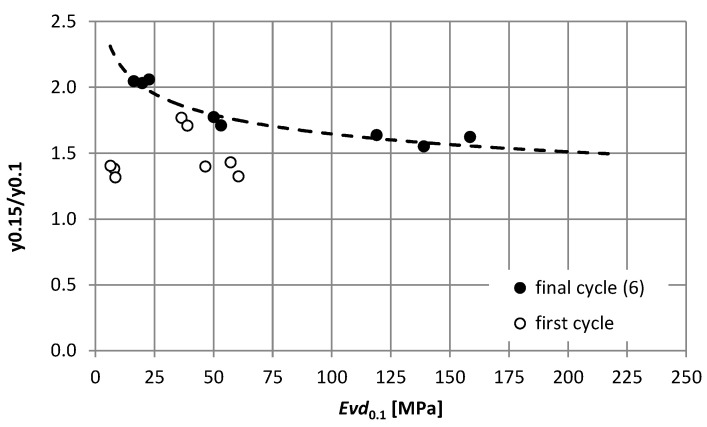
Dependence of the displacement ratios at two load levels (0.1 MPa and 0.15 MPa) on the dynamic modulus determined at the load of 0.1 MPa.

**Figure 8 materials-15-05646-f008:**
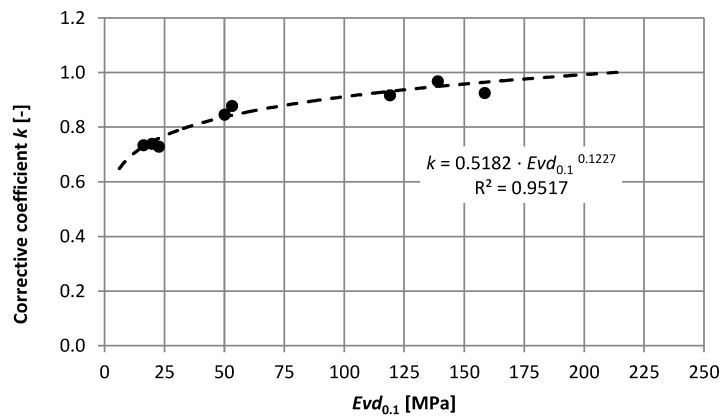
Correction factor for dynamic moduli determined in the test with a plate load of 0.1 MPa.

**Figure 9 materials-15-05646-f009:**
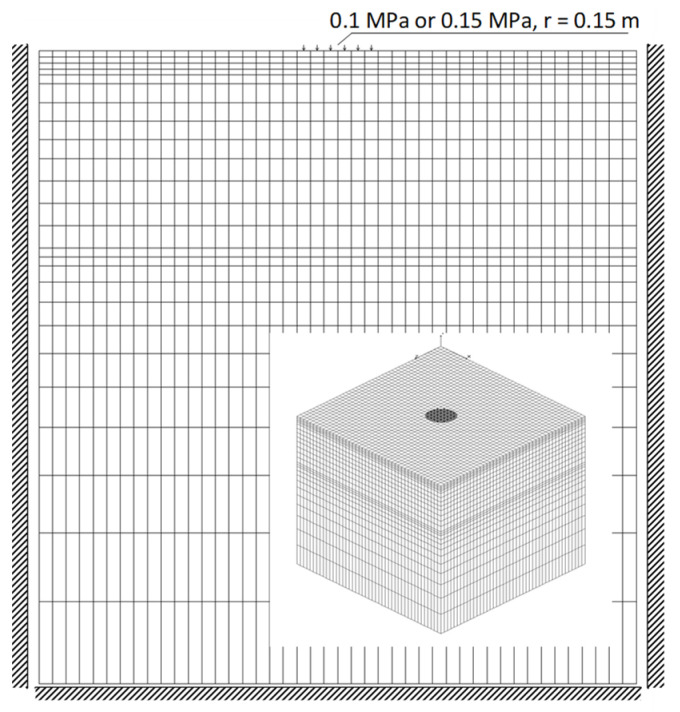
Calculation model of the subgrade layer system.

**Figure 10 materials-15-05646-f010:**
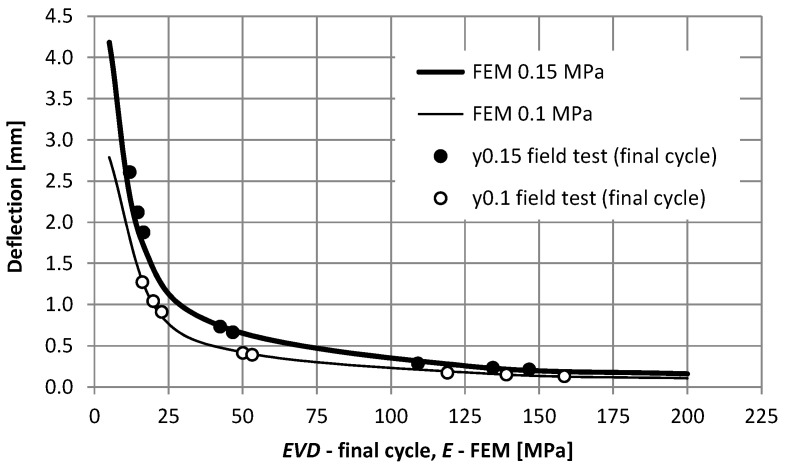
Comparison of the results of field tests and numerical calculations, taking into account moduli and displacements.

**Figure 11 materials-15-05646-f011:**
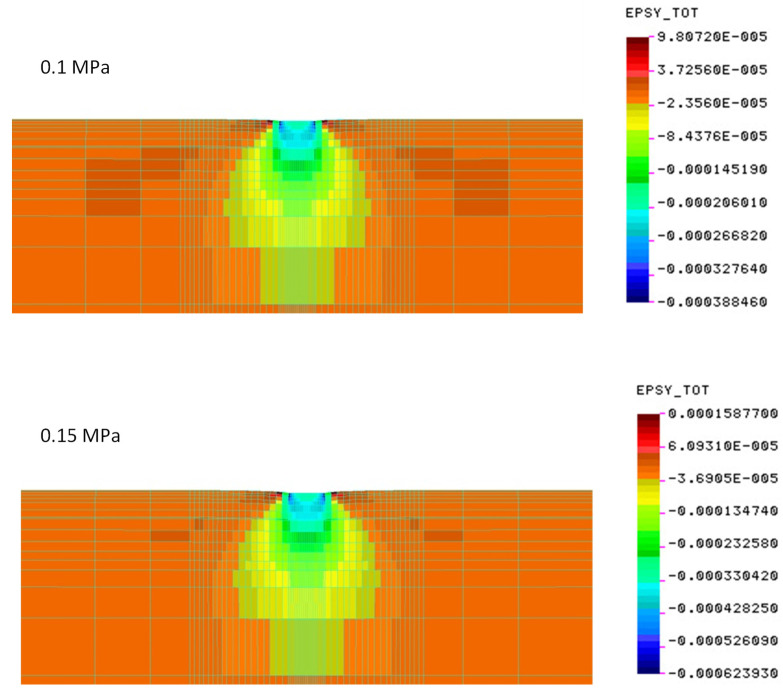
Deformations for crushed stone on a cement-stabilized subgrade—construction No. 1 (deformation scale ×100).

**Figure 12 materials-15-05646-f012:**
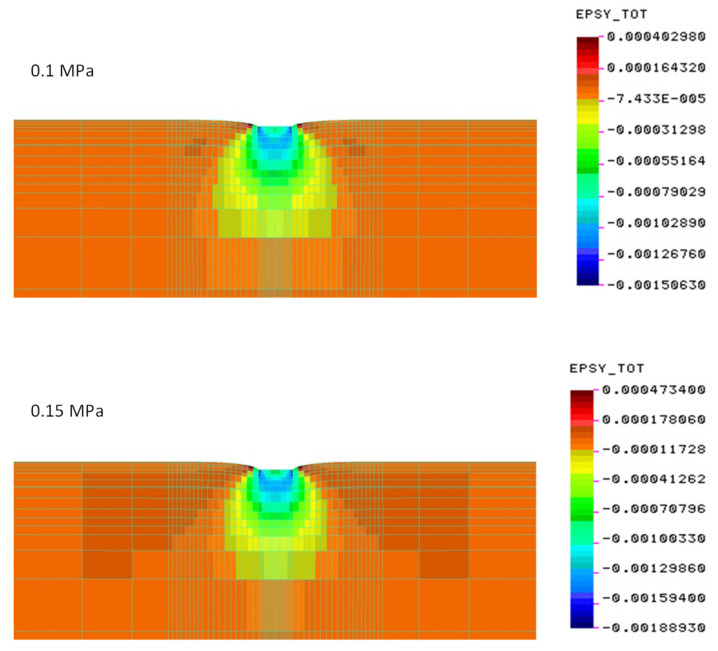
Vertical deformations on the cement-stabilized subgrade—construction No. 2 (deformation scale ×100).

**Figure 13 materials-15-05646-f013:**
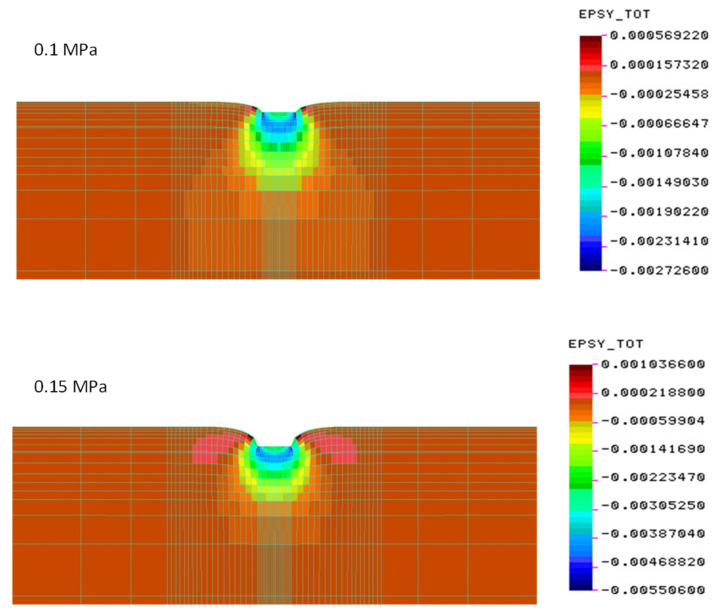
Vertical deformations for the native subgrade—construction No. 3 (deformation scale ×100).

**Figure 14 materials-15-05646-f014:**
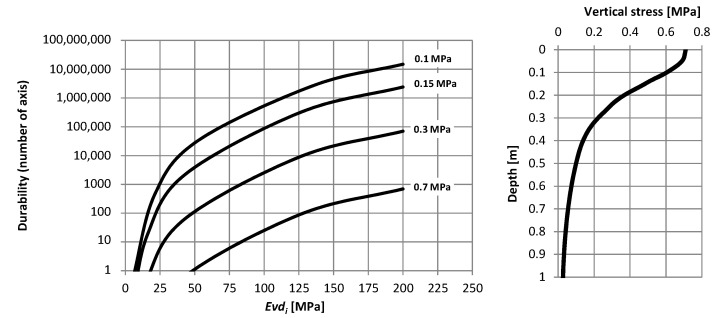
Dependence of durability on modules determined for the tested systems with different replacement modules.

**Figure 15 materials-15-05646-f015:**
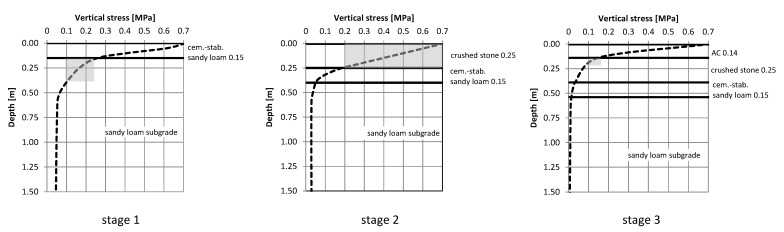
Distribution of vertical stresses in exploited systems, at various stages of construction and in the target system of layers of the pavement structure.

**Figure 16 materials-15-05646-f016:**
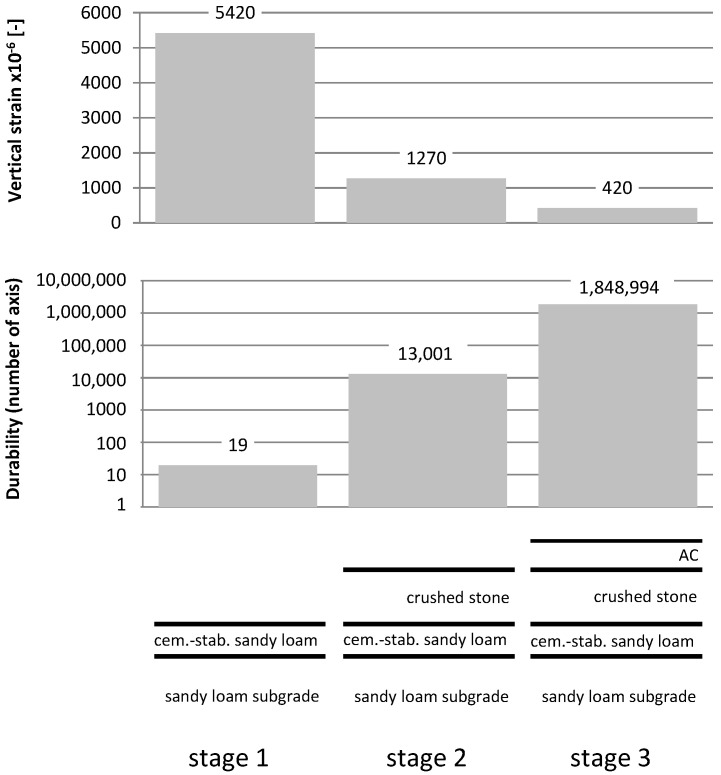
Distribution of vertical stresses in exploited systems at various stages.

## Data Availability

Not applicable.

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
