# Peer review of "Identification of Parameters and Fatigue Life Assessment of the Road Pavement Lower Construction Layers under Heavy Construction Traffic"

_materials, 2022, doi:10.3390/ma15165646_

Round 1

Reviewer 1 Report

The paper entitled “Identification of parameters and fatigue life assessment of the road pavement lower construction layers under heavy construction traffic” is suitable for publication at the materials Journal. The authors carried out numerical calculations (FEM) in the elastic model for layered structures with replacement modules determined in field tests. The authors found that, in the case of low-bearing layer systems, it is necessary to use correction factors for modules determined with a dynamic plate at a load of 0.1 MPa. The reduction of the fatigue life of the lower layers of the pavement and subgrade structure should be taken into account at the design stage of the structure in the future operation of the target system of the pavement structure layers. The methodology, modelling, analysis, and results are correct. The reviewer recommends the paper for publication.

Author Response

The authors thank the reviewer for the positive comment and careful review, which helped improve the manuscript.

Reviewer 2 Report

I have one question asking for revision:

Retreated and rewrite "If the expected load from construction or temporary traffic causes stress of 0.15 MPa in the layers (such stress has been identified in the model as the average occurring in the layer), the modules should be tested under a load of 0.15 MPa or reduction factors for the designated modules in the case of plate testing 0.1 MPa. Otherwise, oversized systems with incorrectly determined layer parameters will degrade faster than the fatigue criteria would indicate. In order to explain these phenomena more precisely and to determine the differences in fatigue life, numerical calculations were carried out for the examined layer systems."

Author Response

The authors have corrected the following paragraph in the document in the review mode. The authors thank the reviewer for the positive comment and careful review, which helped improve the manuscript.

"If the expected load from construction or temporary traffic causes stress of 0.15 MPa in the layers (such stress has been identified in the model as the average occurring in the layer), the modules should be tested under the load of 0.15 MPa or reduction factors, for the designated modules in the case of plate testing 0.1 MPa, should be applied. Otherwise, oversized systems with incorrectly determined layer parameters will degrade faster than the fatigue criteria would indicate. In order to explain these phenomena more precisely and to determine the differences in fatigue life, numerical calculations were carried out for the examined layer systems."

Reviewer 3 Report

The modulus and fatigue life of subgrade and lower layers were tested and analyzed in this study. And FEM was developed to verify the testing results. Its research program and finding have been presented clearly. It can be accepted after minor revision. Here are some comments for improvement: 1) the objectives of this research were not well explained in the Introduction; 2) Section 2 of Research methodology should focus on the methodologies. Listing so much data would make the reader difficult to follow; 3) Section 3 of numerical modeling, input of materials parameters for the involved materials should be provided.; 4) Color pictures for Figure 13 would be better; 5) please add corresponding recommendations for better pavement design, according to this research. What would we get from the discussed tests and modelling?

Author Response

1) the objectives of this research were not well explained in the Introduction;

The authors modified the explanation of the main objectives:

The aim of the article is to show that in the layer systems with equivalent modulus of 80 MPa or lower significant nonlinear phenomena occur, when under stress of 0.15 MPa (identified in the model as the average). Therefore the identification of the equivalent modulus should be conducted under such a stress. Identification under lower stress ( 0.1 MPa) is incorrect and  causes  significant overestimation of the equivalent modulus and consequently overestimation of the fatigue life of the pavement construction.

2) Section 2 of Research methodology should focus on the methodologies. Listing so much data would make the reader difficult to follow;

The authors are aware of the large amount of data, however they find them necessary to explain the main thesis of the article, especially the nonlinear phenomena in the construction layers.

3) Section 3 of numerical modeling, input of materials parameters for the involved materials should be provided.;

Materials parameters have been added to section 3:

The FEM calculations have been conducted for the following moduli obtained in tests: crushed stone on a cement-stabilized subgrade - construction No. 1: 139 MPa (load 0.1 MPa), 134 MPa (load 0.15 MPa);  cement-stabilized subgrade - construction No. 2: 50MPa (load 0.1 MPa), 42 MPa (load 0.15 MPa); native subgrade - construction No. 3: 30 MPa (load 0.1 MPa), 15 MPa (load 0.15 MPa). Poisson ratio of 0.35 has been used in all cases.

4) Color pictures for Figure 13 would be better;

Fig. 11, 12 and have been changed to color.

5) please add corresponding recommendations for better pavement design, according to this research. What would we get from the discussed tests and modelling?

The following text has been added to the final findings:

Special attention should be paid to fatigue life analysis of the low bearing subgrades under temporary or construction traffic. In such cases, the correction factors should be used, when identifying moduli in LWD test under 0.1 MPa load. It is essential to correctly predict the fatigue life in the following stages of the construction and of the final layer system.  

The authors thank the reviewer for the positive comment and careful review, which helped improve the manuscript.

Reviewer 4 Report

The manuscript discusses the parameter identification of three kinds of road surfaces. The numerical analysis well agreed with experimental results, and experimental identification conditions were carefully investigated. Therefore, I only have a few comments.

1) The affiliation should include the city and country.

2) Page 1, line 36 "using a the light" ==> Mistake?

3) Figure 3: The difference in the curves was not shown. Please change the color or line type by the number of cycles, and add explanatory notes.

4) Please describe the boundary conditions at the interface of the layers, such as between crushed stone and sandy loam, in the FEM calculation.

Author Response

1) The affiliation should include the city and country.

The city and country have been added.

2) Page 1, line 36 "using a the light" ==> Mistake?

Page 1, line 36 has been corrected to : light dynamic plate na light weight deflectometer

3) Figure 3: The difference in the curves was not shown. Please change the color or line type by the number of cycles, and add explanatory notes.

Fig. 3 has been changed to color.

4) Please describe the boundary conditions at the interface of the layers, such as between crushed stone and sandy loam, in the FEM calculation.

Full interlayer adhesion has been applied in the model.

The authors have corrected the following paragraph in the document in the review mode. The authors thank the reviewer for the positive comment and careful review, which helped improve the manuscript.